# Preparation, Characterization, In Vitro Release, and Antibacterial Activity of Oregano Essential Oil Chitosan Nanoparticles

**DOI:** 10.3390/foods11233756

**Published:** 2022-11-22

**Authors:** Yuan Ma, Ping Liu, Kunyue Ye, Yezheng He, Siqi Chen, Anqi Yuan, Fang Chen, Wanli Yang

**Affiliations:** Key Laboratory of Food Biotechnology of Sichuan Province, School of Food and Bioengineering, Xihua University, Chengdu 610039, China

**Keywords:** oregano essential oil, chitosan nanoparticles, Box–Behnken design, antibacterial

## Abstract

Essential oils have unique functional properties, but their environmental sensitivity and poor water solubility limit their applications. Therefore, we encapsulated oregano essential oil (OEO) in chitosan nanoparticles (CSNPs) and used tripolyphosphate (TPP) as a cross-linking agent to produce oregano essential oil chitosan nanoparticles (OEO-CSNPs). The optimized conditions obtained using the Box–Behnken design were: a chitosan concentration of 1.63 mg/mL, TPP concentration of 1.27 mg/mL, and OEO concentration of 0.30%. The OEO-CSNPs had a particle size of 182.77 ± 4.83 nm, a polydispersity index (PDI) of 0.26 ± 0.01, a zeta potential of 40.53 ± 0.86 mV, and an encapsulation efficiency of 92.90%. The success of OEO encapsulation was confirmed by Fourier transform infrared spectroscopy (FT-IR) and thermogravimetric analysis (TGA). The scanning electron microscope (SEM) analysis showed that the OEO-CSNPs had a regular distribution and spherical shape. The in vitro release profile at pH = 7.4 showed an initial burst release followed by a sustained release of OEO. The antibacterial activity of OEO before and after encapsulation was measured using the agar disk diffusion method. In conclusion, OEO can be used as an antibacterial agent in future food processing and packaging applications because of its high biological activity and excellent stability when encapsulated.

## 1. Introduction

In recent years, there has been widespread concern regarding the health benefits of functional bioactive ingredients and the rise in negative perceptions of artificial food additives [1]. Plant essential oils (EOs) are commonly found in various plant parts, such as roots, trunks, barks, stems, leaves, flowers, and fruits, and contain numerous terpenes, aromatic, aliphatic, sulfur, and nitrogen compounds. EOs have a variety of biological activities, including those of antioxidants, sterilization, and disinfection [2]. OEO is a natural, transparent, pale-yellow liquid extracted from oregano leaves; it has excellent antibacterial activity (antibacterial, antifungal, and antiviral) [3] and antioxidant properties [4] and is used widely in the medicine and food preservation fields [5]. Its primary components, carbaldehyde, and thymol can cause irreversible damage by affecting the permeability of microbial cell membranes and inhibiting cell respiration and metabolism [6], resulting in a bacteriostatic effect. However, the disadvantages of oregano essential oil, including its high volatility, pungent odor, and insolubility in water, restrict its application [7]. Nanoencapsulation is a promising strategy for increasing the value of OEO by extending their validity period, enhancing their physicochemical stability, and achieving controlled release despite these obstacles [8].

The delivery of bioactive compounds found in essential oils through nanoparticles not only protects the bioactive compounds of essential oils from potential degradation caused by direct exposure to different environmental factors (light, heat, pH, humidity, and oxygen) [9]. In addition, the unpleasant odor of essential oils is masked, their release is regulated, their solubility and stability are enhanced, and they have an inherent antibacterial effect [10] that enhances their effectiveness [9]. Studies have shown that nanoparticle delivery systems can deliver essential oils to the surface of bacterial cell membranes and enhance their absorption, whereas pure essential oils (low water solubility) do not readily contact the cell membrane [11]. Chitosan nanoparticles containing essential oils had antibacterial activity compared to essential oils alone [11].

Chitosan (CS) is a naturally occurring cationic alkaline polysaccharide derived from the deacetylation of chitin, which is abundant in shrimp, crabs, silkworms, algae, and other substances [12]. Due to its biocompatibility, biodegradability, near-non-toxicity, and inexpensive production [13], the use of CS in pharmaceutical and food applications has increased dramatically in recent years [14]. Chitosan is the most studied and suitable nanocarrier for essential oil delivery. Chitosan nanoparticles can enhance the thermal stability of essential oils, preserve their phenolic content, ensure prolonged release profiles, and enhance their antioxidant and antibacterial activities for numerous applications. Studies have shown that encapsulation of oregano essential oil in nanoparticles derived from chitosan significantly enhanced its antifungal activity [15]. The most classic method for preparing chitosan nanoparticles is ionic cross-linking. Chitosan can also interact electrostatically with certain polyanionic substances in addition to being combined with negatively charged macromolecules, thereby further forming nanoparticles as a result of intramolecular and exo-molecular cross-linking that occurs in the presence of polyanions [16]. The method has the advantages of simple operation, a high drug-loading capacity, the absence of organic solvents, the avoidance of toxic and side-effects caused by the cross-linking agent, and the capacity to maintain the integrity and biological activity of the nanostructure after freeze-drying and storage [17].

The essential oils of cinnamon [18], clove [19], and Heracleum persicum fruit [20] were encapsulated in chitosan nanoparticles, and their properties and antibacterial activities were studied, laying the foundation for their application. The *genus Pseudomonas* comprises a heterogeneous group of *Pseudomonadaceae* microorganisms. Members of the *genus Pseudomonas* are frequently associated with the degradation and spoilage of a wide variety of foods derived from plants or animals, as their very simple nutritional requirements and metabolic diversity enable them to thrive in a variety of environments [21]. Few reports have described the encapsulation of OEO into chitosan nanoparticles, and even fewer have optimized their preparation process, characterized and evaluated the sustained release properties of the optimized nanoparticles, or assessed their *Pseudomonas*-inhibiting activity. These studies contribute to solving the poor stability of oregano essential oil while improving its environment tolerance and slow release, thereby substantially expanding its application in preservation and food preservation.

In this study, to determine the optimal conditions for preparing nanoparticles with the highest encapsulation efficiency and the smallest particle size, OEO was encapsulated in CSNPs using ionic cross-linking of three independent variables and TTP as a cross-linking agent (CS concentration, three effects of TPP concentration, and the addition amount of OEO). FT-IR, TGA, and SEM were used to study the encapsulation of optimally produced OEO-CSNPs. In addition, the in vitro release abilities of OEO-CSNPs were examined using the three *Pseudomonas* strains: the bacteriostatic ability of bacteria (*Pseudomonas fluorescens* (MN578148.1), *Pseudomonas gessardii* (MN069032.1), and *Pseudomonas jessenii* (MN758767.1).

## 2. Materials and Methods

### 2.1. Chemicals and Reagents

OEO was purchased from Ji’an Huashuo Fragrance Oil Co., Ltd. (Jian, China). CS (degree of deacetylation 90%) was provided by Shanghai Yuanye Bio-Technology Co., Ltd. (Shanghai, China). TPP was purchased from Shandong Yousu Chemical Technology Co., Ltd. (Linyi, China). Sodium hydroxide, disodium hydrogen phosphate, dodecahydrate, and sodium chloride were purchased from Chengdu Kelong Chemical Co., Ltd. (Chengdu, China). Nutrient agar was purchased from Beijing Auboxing Biotechnology Co., Ltd. (Beijing, China). All chemicals were used as received without further purification. All of the reagents and solvents listed above, and others, were of analytical grade.

### 2.2. Preparation of OEO-CSNPs

The ion cross-linking method was used to prepare nanoparticles, as reported by Zeinab et al. [22]. To obtain a chitosan solution, an appropriate quantity of chitosan powder was weighed (BSA224S-CW, Sartorius, Beijing, China), dissolved in 1% acetic acid solution at room temperature, and stirred (RH digital, Ai Card Instrument Equipment Co., Ltd., Guangzhou, China) at 600 r/min for 12 h. Before filtering through a 0.45 μm microporous membrane, the pH was adjusted to 4 with a 5 mol/L sodium hydroxide solution. The essential oil emulsifier Tween-80 was added, and the mixture was stirred and thoroughly mixed. Oregano essential oil was added to the filtered chitosan solution, and the mixture was stirred at 600 r/min for 60 min while remaining at room temperature. To obtain the OEO-CSNPs suspension, chitosan and TPP solution were combined (TPP was dissolved in ultrapure water and thoroughly dissolved before filtering through a 0.45 μm microporous membrane) at a volume ratio of 5:2, and the mixture was gently stirred for 60 min. The inner layer of an ultrafiltration centrifuge tube (Millipore, MA, USA) was removed after centrifugation (5810 R, Eppendorf Company, Hamburg, Germany) at 8000 r/min for 30 min, washed several times with ultrapure water, and then ultrasonicated (KH3200E, Jintan Medical Center, Changzhou, China). OEO-CSNPs were dispersed (Instrument Factory, China) for 5 min, pre-frozen at −76 °C (ULTS1386, Thermo Fisher, Waltham, MA, USA) for 3 h, and then vacuum-lyophilized (FDU-1200, Shanghai Ailang Instrument Co., Ltd., Shanghai, China). It was stored at 4 °C until use. The formation process of OEO-CSNPs is shown in Figure 1.

### 2.3. Optimization of the Preparation of OEO-CSNPs Using BBD

Various crucial process parameters affecting the encapsulation efficiency (%EE) and particle size of oregano essential oil chitosan nanoparticles were investigated in the preliminary experiments. The encapsulation efficiency and particle size of OEO-CSNPs were significantly affected by the CS concentration, TPP concentration, and the amount of OEO added (with CS solution as the control, *v*/*v*). The volume ratio, stirring speed, and stirring time are fixed parameters. Utilizing the BBD design principle, a 3-factor, 3-level complete factorial design was implemented. The coded and actual values of the independent and dependent variables are shown in Table 1. By analyzing the effect of independent variables (A-CS concentration, B-TPP concentration, and C-essential oil addition amount) on dependent variables (*Y*_1_-%EE, *Y*_2_-Size), the Design-expert software was used to optimize oregano essential oil chitosan nanoparticles (Design-expert, 8.0.6). Fitting oregano essential oil chitosan nanoparticles with the second-order polynomial equation revealed that the independent variable was correlated with the dependent variable, and the general form was (1):(1)$$Y_{i}=b_{0}+b_{1}A+b_{2}Β+b_{3}C+b_{12}AΒ+b_{13}AC+b_{23}ΒC+b_{11}A^{2}+b_{22}{Β}^{2}+b_{33}C^{2}$$
where *Y_i_* represents the observed dependent variable response for each factor level, *b_0_* is the intercept, and *b_1_* to *b_33_* are the regression coefficients obtained from the observed response values during the experiment. The terms A, B, and *C* are independent variables, whereas (AB, A*C*, and B*C*) and (A^2^, B^2^, and *C*^2^) are interaction and quadratic effects, respectively [23]. The experimental design included a total of 17 experiments. For each of the 17 experiments, the midpoint experiment was repeated 5 times to ensure reliability. Analysis of variance (ANOVA) was used to verify the mathematical model fitting the data. The sufficiency of the second-order polynomial model was determined by examining the significance of the regression coefficients, the coefficient of determination (R^2^), and the absence of fit for each investigated response. Three-dimensional response surface plots were used to show the interaction and influence of variables on responses [24]. Following the development and validation of the model, the optimal formulation with the highest encapsulation efficiency and smallest particle size was selected, and evaluation experiments were conducted, including characterization and physicochemical properties.

### 2.4. Maximum Absorption Wavelength and Standard Curve of OEO

The OEO was diluted with absolute ethanol to a predetermined concentration, and then a full-wavelength scan in the 200–800 nm wavelength range was conducted with a Shimadzu dual-beam UV-Vis spectrophotometer (UV-1900, Shimadzu, Kyoto, Japan). The maximum absorption wavelength of OEO was determined using ethanol as a blank control. OEO has a maximum absorption peak at the wavelength of 276 nm, which can be used to determine the entrapment rate of OEO-CSNPs, according to experimental findings.

At the wavelength of maximum OEO absorption, OEO was diluted in absolute ethanol to various concentration gradients (0.005, 0.01, 0.02, 0.04, and 0.06 mg/mL), and the absorbance was measured at each concentration. A standard curve was drawn with the equation y = 16.984x + 0.0085, R^2^ = 0.999 for the blank control.

### 2.5. Determination of Encapsulation Efficiency and Loading Capacity

A MercMillipore ultrafiltration centrifuge tube (UFC901096, Shanghai Bitai Biotechnology Co., Ltd., Shanghai, China) was used to centrifuge the OEO-CSNPs at high speed (8000 r/min) for 30 min at 4 °C [25]. The free essential oil was then transferred to the outer centrifuge tube. An ultraviolet spectrophotometer (UV2400, Shanghai Sunny Hengping Scientific Instruments, Shanghai, China) set to a detection wavelength of 276 nm was used to analyze the supernatant in the outer centrifuge tube, yielding information about the concentration of free essential oil. From the inner centrifuge tube, nanoparticles were extracted, separated, washed, freeze-dried, and weighed. The *EE* and *DL* of *OEO-CSNPs* were calculated using Formulas (2) and (3), respectively:(2)$$EE(\%)=\frac{Total\;amount\;of\;OEO-Free\;amount\;of\;OEO}{Total\;amount\;of\;OEO}\times 100$$
(3)$$DL(\%)=\frac{Total\;amount\;of\;OEO-Free\;amount\;of\;OEO}{Weight\;of\;nanoparticles\;after\;freeze\;drying\;OEO}\times 100$$

### 2.6. Particle Size, PDI, and Zeta Potential

The particle size, polydispersity coefficient (PDI), and zeta potential of nanoparticles were measured by a Zeta Potentiometer-Laser Particle Size Analyzer (ZEN-3600) from Malvern, UK, using the method of Kalam et al. [26]. A certain quantity of prepared OEO-CSNPs was dispersed in ultrapure water. The fixed angle of scattered light was 90°, and the measured temperature was 25 ± 1 °C.

### 2.7. Scanning Electron Microscopy (SEM)

The microscopic morphology of freeze-dried oregano essential oil chitosan nanoparticles was examined using a German ZEISS Gemini SEM 300 scanning electron microscope. The sample should be secured to the sample stage with double-sided tape, and gold should be observed and sprayed [27]. 

### 2.8. Fourier Transformed Infrared Spectroscopy (FT-IR)

The ATR method was used to analyze the liquids, whereas the tableting method was used to analyze the powder samples (after grinding with dry KBr (1/100, m/m), tableting was performed). At room temperature, the tablets were scanned on a Fourier transform infrared spectrometer from 4000 cm^−1^ to 400 cm^−1^ for 32 consecutive scans with a resolution of 4 cm^−1^ (Nicolet AVATAR 380, Thermo Fisher Scientific Inc., Waltham, MA, USA).

### 2.9. Thermal Characterization by Thermogravimetric Analysis (TGA)

A thermogravimetric analyzer was used to assess the thermal stability of the nanoparticles in a 3–5 mg sample (DTG-60, Shimadzu, Japan). The tests were performed in an atmosphere of nitrogen at a gas flow rate of 30 mL/min and a temperature increase heating rate of 10 °C/min from 30 to 600 °C [28].

### 2.10. In Vitro Release Studies 

A mass of 20 mg of nanoparticles was placed in 30 mL of phosphate buffer (pH 7.4) and shaken (ZWY-100II, Shanghai Zhicheng Analytical Instrument Manufacturing, Shanghai, China) at 37 ± 0.5 at 120 rpm, according to the reported method [29]. Aliquots of 3 mL were removed and replaced with an equal volume of fresh dissolution medium at a specific time intervals (0.25, 0.50, 0.75, 1.0, 1.5, 2.0, 3.0, 5.0, and 24.0 h, once a day for 8 days). UV-Vis spectrophotometry at 276 nm was used to examine the samples against a PBS reference solution. All experiments were conducted at 25 °C, and all measurements were repeated three times.
(4)$$\mathrm{Cumulative}\;\mathrm{release}\;\mathrm{percentage}=\sum_{t\;=\;0}^{t}\frac{M_{t}}{M_{0}}\times 100$$
where M_t_ is the cumulative amount of OEO released to each sampling time point, t is the release time of OEO-loaded, and M_0_ is the initial weight of OEO-loaded in the sample.

### 2.11. Determination of Antimicrobial Activity

First, OEO-CSNPs were diluted with sterile water at a ratio of 1:100 (*w*/*v*), according to the concentrations of chitosan nanoparticles and essential oil in the particle solution. *Pseudomonas fluorescens* (MN578148.1), *Pseudomonas gessardii* (MN069032.1), and *Pseudomonas jessenii* (MN758767.1) were the three most common strains found in the previous screening, as reported by Esmaeili et al. [30]. The filter paper diffusion method was used to evaluate antibacterial activity after cultivating the selected strains to the logarithmic growth phase. A sterile spreader bar was used to evenly distribute the bacterial suspension on nutrient agar (LB) plates at a concentration of 10^6–^10^7^ CFU/mL (0.5 McFarland turbidity standard). Subsequently, filter paper pieces (6 mm in diameter) were impregnated with 20 μL of the above sample solution and placed on the plate. Controls consisted of sterile water and absolute ethanol. All dishes were then incubated at 30 °C for 24 h. The size of the inhibition zone should be observed.

### 2.12. Statistical Analysis

All experiments were repeated three times. Data were processed and analyzed in Excel 2010 and SPSS 24 software and graphs were created in Origin 8.5, respectively. Analysis of variance (ANOVA) and significant differences (Duncan’s multiple comparison procedure) were performed (*p* < 0.05).

## 3. Results

### 3.1. BBD Experimental Results and Analysis

The benefit of BBD is that it excludes combinations of all factors that simultaneously appear at the highest or lowest levels, thereby preventing experiments conducted under extreme conditions from yielding unsatisfactory results [23]. BBD analysis was used to examine the main effect, interaction effect, and quadratic effect of three independent variables (A, B, and *C*) on the response value (*Y*_1_ and *Y*_2_). The 17 obtained test results are shown in Table 2. The particle size values varied significantly, from 684.37 nm to 186.63 nm, and the encapsulation efficiencies ranged from 77.54% to 92.18%. This result is similar to previous results where chitosan nanoparticles and different amounts of *Carum copticum* EO were encapsulated with their average size ranging from 721.0 nm to 236.0 nm [30]. Wu et al. [31] studied the similar encapsulation rate of essential oils when they were encapsulated into nanoparticles. It was determined, through analysis of experimental data in Design-Expert V8.0.6 software, that the encapsulation efficiency (*Y*_1_) and particle size (*Y*_2_) are related to CS concentration (A), TPP concentration (B), and the amount of OEO added (*C*). The equation of the multinomial regression model is as follows:(5)$$\begin{array}{cc} Y_{1} & =92.59+0.39A-3.75Β+2.05C-0.50AΒ+0.67AC+1.19ΒC\\ & -3.73A^{2}-3.62{Β}^{2}-3.73C^{2} \end{array}$$
(6)$$\begin{array}{cc} Y_{2} & =202.13-95.66A+22.47Β+20.15C+35.56AΒ-43.37AC\\ & +113.80ΒC+165.92A^{2}+62.77{Β}^{2}+156.13C^{2} \end{array}$$

Positive values in the regression equation for response values indicate synergistic optimization effects, whereas negative values indicate antagonistic effects between factors and response values [26].

Significant parameters influencing encapsulation efficiency and particle size were identified using ANOVA (Table 3). All of the obtained regression models were highly significant (*p* < 0.01), and the lack of fit was not significant (*p* > 0.05), indicating that the models were effective and had high predictive values. The coefficients of determination R^2^ for packaging efficiency and particle size were 0.9915 and 0.9864, respectively, and the adjusted R^2^ was 0.9805 and 0.9690, respectively. Generally, the higher the R^2^ value, the better a quadratic polynomial model can represent the fitting effect of the quadratic model on the data under experimental conditions. The *p*-value of each coefficient indicates its significance. Values (*p* > 0.05) indicate significant variables. Optimizing the experimental factors facilitates the development of nanoparticles with high encapsulation efficiency and small particle size (Table 3).

The three-dimensional surface representation of the experiment is shown in Figure 2. The curved surface shows that the two factors interacted strongly. Figure 2a–c show that, when one was held constant, the encapsulation efficiency was affected by the other two factors. When the CS concentration was 1.5~1.8 mg/mL and the TPP concentration was 1~1.4 mg/mL, increasing the amount of essential oil added from 0.25% to 0.40% resulted in a higher EE%. Due to chitosan and TPP, the EE% decreased as the chitosan concentration, TPP concentration, and essential oil addition amount increased continuously, and the EE% decreased, due to chitosan and TPP. The saturation of the binding sites in the medium and increasing the amount of essential oil can reduce the EE%. Similar results have been obtained in encapsulation of eugenol [32] and *Satureja hortensis* essential oil [33] in particles. The particle size decreased from 1 mg/mL to approximately 1.8 mg/mL of CS concentration, as shown in Figure 2d–f, before increasing at approximately 1.8 mg/mL of CS concentration. Molecular weight and chitosan concentration can be adjusted to change particle size: higher chitosan concentrations are more conducive to small particle formation [17]. However, because many CS molecules are involved in the cross-linking of individual particles at higher CS concentrations, the intermolecular hydrogen bond attraction is stronger, and the electrostatic repulsion is insufficient, leading to the formation of larger particles upon the addition of TPP. This phenomenon can be explained by the fact that as the CS concentration increases, intermolecular distance decreases, resulting in an increase in intermolecular crosslinks between CS molecules. Reduced cross-linking density between CS and TPP caused particle aggregation and the formation of larger particles [34]. The particle size begins to increase when the TPP concentration exceeds 1.5. Encapsulation of herbal galactagogue extract obtained comparable results [17]. A possible reason is that an increase in TPP concentration results in greater electrostatic interaction between it and chitosan, which increases particle size [17].

The maximum encapsulation efficiency and smallest particle size were achieved at a CS concentration of 1.63 mg/mL and TPP concentration of 1.27 mg/mL, respectively, using the regression equation to determine the ideal conditions for optimally preparing OEO-CSNPs. The amount of added essential oil was 0.30%. The model validation was confirmed by preparing nanoparticles in duplicate under these conditions three times and comparing the results to the values predicted by the model. The experimental results are shown in Table 4. The prepared nanoparticles had an average particle size of 182.77 nm, encapsulation efficiency of 92.90%, and drug loading of 27.63%. The errors between predicted and experimental values were all less than 5%, indicating that the model can predict optimal conditions within the horizontal domain defined by the independent variables. Therefore, the response surface regression model is capable of accurately predicting nanoparticle particle size and encapsulation efficiency [35], and compared to the study by Cai et al. [36], nanoparticles with high EE and LC and low particle size were obtained after optimizing the preparation conditions by BBD experiments.

### 3.2. Characterization of OEO-CSNPs

Essential evaluation criteria for nanoparticle performance include particle size, zeta potential, and PDI. The CS concentration was 1.63 mg/m, under optimal preparation conditions, the TTP concentration was 1.27 mg/mL, the essential oil concentration was 0.30%, the pH was 4.0, and the stirring time was 60 min. Figure 3 shows that the average particle size of OEO-CSNPs was 182.77 ± 4.83 nm, with a typical size distribution curve, and the size distribution was narrow (polydispersity coefficient PDI was 0.26 ± 0.01, PDI was correlated with molecular stability). Mohammadi et al. [37] studied the average size of 100–190 nm after the *Cinnamomum zeylanicum* essential oil was encapsulated into nanoparticles. Cai et al. [36] found that the PDI values of chitosan nanoparticles with *Ocimum basilicum* EO ranged from 0.222 to 0.425. The smaller the value, the more effectively and uniformly nanoparticles are dispersed throughout the solution [38]. The electrostatic interaction between the surface charges of the particles will result in particle aggregation, dispersion, and flocculation. Therefore, to detect the size of the charge and then evaluate the stability of the nanoparticles, the zeta potential was determined. If the zeta potential value of a suspension is higher than +30 mV, it is considered physically stable [39]. The zeta potential of the prepared OEO-CSNPs was +40.53 ± 0.86 mV, indicating that they were remarkably stable (Figure 4). López-Meneses et al. showed that the zeta potential of chitosan nanoparticles after the addition of *Schinus molle* EO was +40.2 ± 7.5 mV [39]. Hesami et al. [40] found that the zeta potential value of CNPs containing *Chelidonium majus* EO was in the range of +26.46 to +33.1 mV. Due to the cationic nature of CS and the presence of residues that do not interact with TPP molecules, the zeta potential was positive. These amino groups inhibit the adsorption of anions, resulting in the formation of stable nanoparticles through the maintenance of a thick electric double layer [23].

### 3.3. SEM Analysis

The morphology of the nanoparticles was observed by a scanning electron microscope. The SEM images of CSNPs (A) and OEO-CSNPs (B) in Figure 5 show that the nanoparticles exhibited a spherical regular distribution and stability during the preparation process. The shape of chitosan nanoparticles was the same before and after the addition of OEO, and Kbaa et al. [35] also found that *Cymbopogon citratus* EO incorporation did not alter the particle shape. Comparing the size of chitosan nanoparticles before and after OEO addition reveals that the new particles were marginally larger. However, Sotelo-Boyas et al. found [41] that the nanoparticle size differed depending on the preparation method and experimental conditions and that the increase in nanoparticle size was associated with CEO adsorption or interaction with each CSNP component [39]. 

### 3.4. FT-IR Analysis

Figure 6 shows the FT−IR spectra of chitosan powder (a), CSNPs (b), OEO (c), and OEO−CSNPs (d).

The overlap of the -OH stretching vibration absorption peak of hydrogen bonds and -NH_2_ stretching vibration absorption peak at 3413 cm^−1^ indicates the formation of the stretching vibration. Chitosan contains numerous distinct hydrogen bonds: a broad peak range at 2809 cm^−1^ (-CH stretching), and three characteristic absorption peaks of amino polysaccharides: 1634 cm^−1^ (amide I, C=O stretching), 1600 cm^−1^ (amide II, N-H bending), and 1350 cm^−1^ (amide III, N-H bend) (Figure 6a) [30]. As shown in Figure 6c, the cross-linking of chitosan and TPP is expected to shift some of the amide group-related peaks, with the peak at 1634 cm^−1^ shifting to 1630 cm^−1^ and the peak at 1600 cm^−1^ shifting to 1591 cm^−1^, confirming the occurrence of ionic cross-linking reaction [30]. Similar results were obtained by Matshetshe et al. [29] studying the interaction between chitosan and TPP. In addition, the appearance of 1249 cm^−1^ (C-O-C stretching), 1456 cm^−1^ (amide II), and 1733 cm^−1^ (C=O stretching) may be attributed to electrostatic interaction [15]. The absorption peaks of oregano essential oil (Figure 6d) are 2959 cm^−1^ (-OCH_2_ stretching), 1589 cm^−1^ (N-H bending), 1458 cm^−1^ (CH_2_, bending), 1254 cm^−1^ (C-O-C stretching), 1117 cm^−1^ (C-O-C stretching), and 937 cm^−1^ (-CH bending) [42]. The above characteristic absorption peaks appear at the same wavenumber, and the OEO significantly increases the intensity of the C-H stretch peak at 2867–2955 cm^−1^, as shown in Figure 3e at 2923 cm^−1^. Strong C-H stretching vibration peaks indicate that the OEO ester functional groups are encapsulated within the nanoparticles [36]. These findings are in agreement with the results of Esmaeili et al. [30], Seyed et al. [34], and Hesami et al. [40]. The results showed that OEO was encapsulated in CSNPs without changing its structure or function, indicating that its antibacterial and antioxidant properties remained intact.

### 3.5. TGA Analysis

TGA analysis is a thermal analysis technique that measures the relationship between the mass of the test sample and the change in temperature at a specific program-controlled temperature [36]. The TGA thermograms in Figure 7 revealed mass loss as a function of temperature for the chitosan powder (a), CSNPs (b), OEO (c), and OEO−CSNPs (d). OEO begins to decompose at approximately 68 °C in the nitrogen environment and loses approximately 92.54% of its mass by 170 °C, which fully reflects the volatile characteristics of essential oil core material [43]. This is similar to the results of Seyed et al. [42] studying the weight loss of OEO. In contrast, the wall material has higher stability. At 243 °C, chitosan undergoes its initial degradation, resulting in a mass loss of approximately 8% due to the evaporation of adsorbed water associated with the hydrophilic group of the polymer. Dehydration of the hydro-glycosidic ring, deacetylation, depolymerization, and decomposition of acetylated chitosan units resulted in the second degradation at 327 °C [30]. CSNPs exhibited different thermal properties than chitosan due to the reaction between CS and TPP. The initial degradation of CSNPs occurred at 230 °C, whereas the second occurred at 385 °C, thereby increasing the degradation temperature of CS. The CEO−CSNPs were degraded at 230 °C and 400 °C, respectively. In conclusion, the chitosan and TPP cross-linking reaction occurred, resulting in the encapsulation of OEO. The OEO−CSNPs shifted the first OEO degradation event from 68 °C to 230 °C and the second from 170 °C to 400 °C, resulting in increased stability [44]. Hosseini et al. [33] and Esmaeili et al. [30] also reported similar findings.

### 3.6. In Vitro Release Analysis

Figure 8 shows that OEO-CSNPs can be released continuously for several days, with a biphasic release pattern consisting of a burst release and a subsequent slow release. This corroborates the findings of Sotelo-Boyas et al. [41]. The burst release exhibited a release amount of 41.27 ± 0.56% at the 5th hour, and the subsequent 8 days of continuous release resulted in 82.73 ± 1.53%. Rapid dissolution of the initial burst could have caused the instant burst of molecules of essential oil molecules adsorbed to the surface of the polymer [35]. Studies suggest that polymer degradation is the primary mechanism by which CSNPs release essential oils into the external environment [45]. The slower release phase is caused by the diffusion of encapsulated essential oil molecules from the nanoparticle core to the dissolution medium through interconnected pores and channels in the polymer matrix. When combined with the hydrophobic behavior of essential oils, they exhibit a slower water absorption rate, thereby decreasing the polymer degradation rate, which is similar to other results concerning hydrophobic materials [46]. Therefore, the chitosan nanoparticles found in oregano essential oil lengthen its duration of action and allow for sustained release [47]. Esmaeili et al. [30] and Yin et al. [48] reported that encapsulation effectively reduced the loss and volatilization of essential oils and their migration rate into the external environment. Encapsulating the essential oil and using it as an antibacterial agent ensures that the initial stage is not too slow to achieve a suitable antibacterial concentration and that the slow stage can maintain activity for an extended time. In contrast, the essential oil becomes ineffective if the initial phase is too rapid [49].

### 3.7. Antibacterial Activity Analysis

The antibacterial activity of CSNPs before and after OEO encapsulation was investigated using the agar disk diffusion method (Table 5). Chitosan nanoparticles exhibited antibacterial activity against the tested strains, and when observed under a microscope, chitosan nanoparticles had a spherical microstructure, which caused the surface of CSNPs to have stronger positive charges, as is typical with bacteria. Additionally, due to the enhanced charge interactions and increased surface area, chitosan can be tightly absorbed into the bacterial surface, destroying the structural integrity of the cell membrane and resulting in intracellular leakage and subsequent cell death. This may be due to the ionic reaction of chitosan with anionic groups on the bacterial cell surface, which changes the bacterial surface morphology, increases membrane permeability, and ultimately results in intracellular substance leakage and cell death [50]. The primary components of OEO, thymol, and carvacrol exhibited antibacterial activity against all three Pseudomonas strains attributed [6]. The OEO encapsulated by CSNPs showed the highest antibacterial activity, with average inhibition zones of 6.86 ± 0.05, 8.30 ± 0.05, and 9.35 ± 0.04 mm against *P. fluorescens*, *P. gessardii*, and *P. jessenii*, respectively. Yilmaz et al. [51] reported that chitosan could enhance the antibacterial ability of essential oils. The antibacterial activity varied between various strains. The difference is a manifestation of the strain specificity. The results showed that encapsulation of OEO in CSNPs improved its antibacterial activity, which is consistent with Feyzioglu et al. [1], Cui et al. [6], and Yilmaz et al. [51].

## 4. Conclusions

In this experiment, the conditions for the preparation of OEO-CSNPs were optimized and characterized using three factors and three concentrations of BBD by an ionic cross-linking method. OEO-CSNPs with high EE and small average particle size and stability were obtained. The FI-TR and TGA results showed that OEO was encapsulated successfully in nanoparticles with good thermal stability. OEO-CSNPs showed a significant sustained release effect and enhanced the antibacterial activity of OEO against Pseudomonas. In conclusion, encapsulating OEO in CSNPs showed its antibacterial utility and active packaging potential application. It serves as a reference for the future development of bactericidal preservatives derived from plant essential oils on green aquatic products, fruits, and vegetables.

## Figures and Tables

**Figure 1 foods-11-03756-f001:**
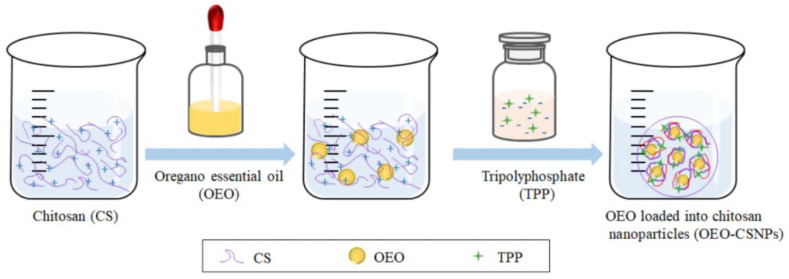
The formation process of OEO-CSNPs.

**Figure 2 foods-11-03756-f002:**
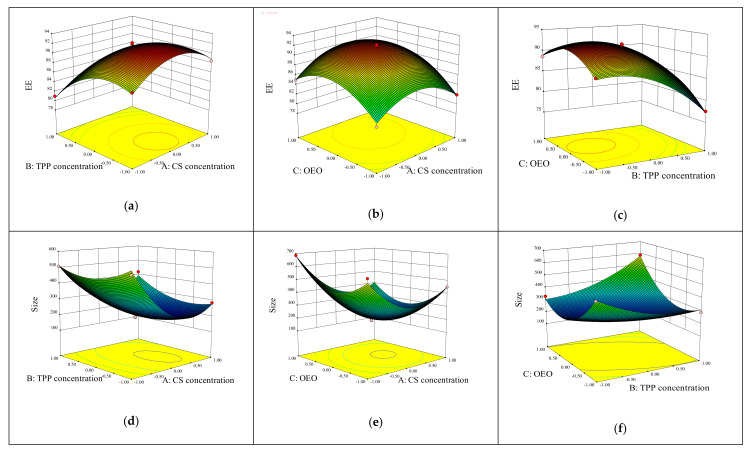
The three-dimensional surface representation of the experiment. Plots (**a**–**c**) and (**d**–**f**) show the effects of CS concentration (A), TPP concentration (B), and OEO addition (C) on the EE% and particle size of OEO-CSNPs, respectively.

**Figure 3 foods-11-03756-f003:**
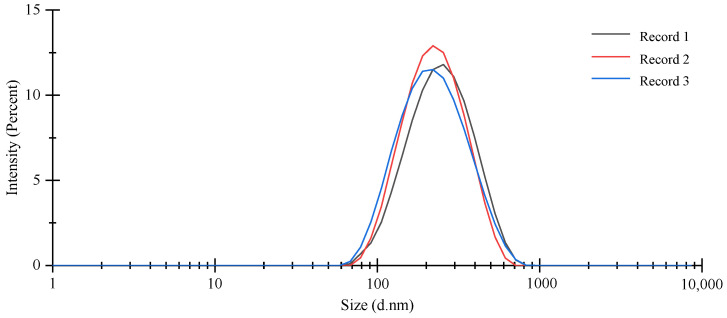
Particle size of OEO-CSNPs. Records 1−3 represent the first few repetitions of the experiment.

**Figure 4 foods-11-03756-f004:**
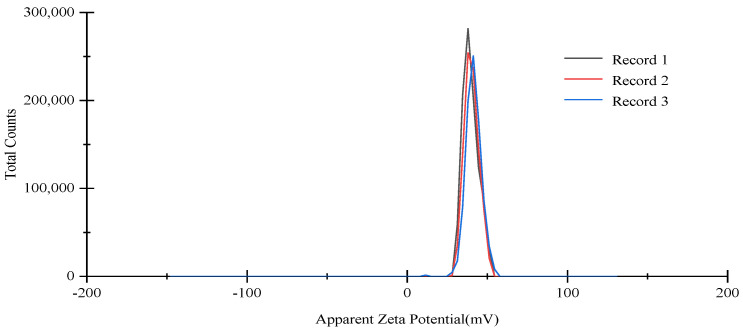
OEO-CSNPs zeta potential. Records 1−3 represent the first few repetitions of the experiment.

**Figure 5 foods-11-03756-f005:**
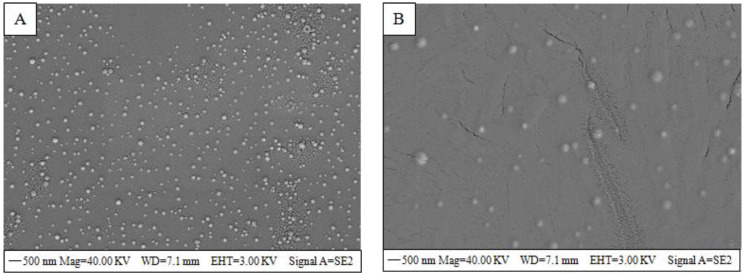
SEM images of (**A**) CSNPs and (**B**) OEO−CSNPs.

**Figure 6 foods-11-03756-f006:**
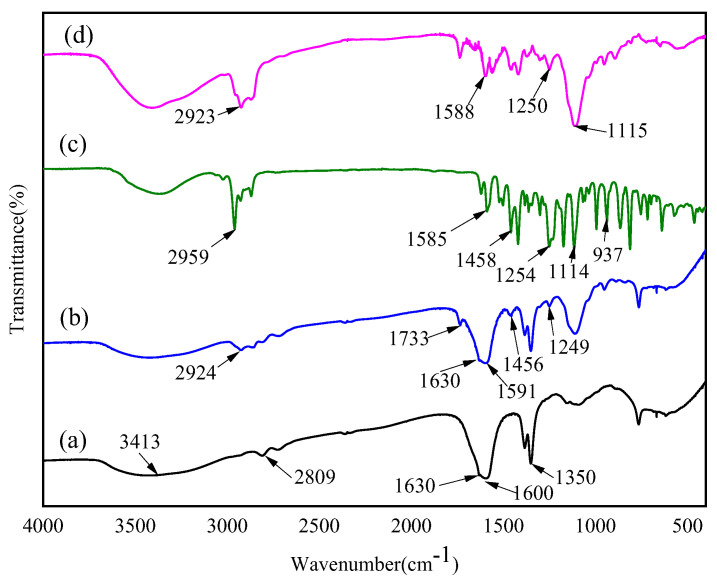
FT-IR spectra of chitosan powder, (**a**), CSNPs (**b**), pure OEO (**c**), and OEO-CSNPs (**d**).

**Figure 7 foods-11-03756-f007:**
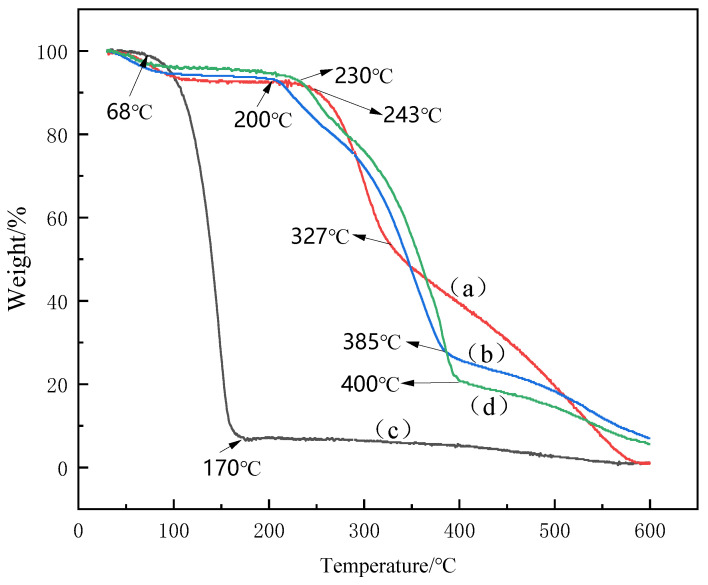
TGA of chitosan powder (**a**), CSNPs (**b**), pure OEO (**c**), and OEO- CSNPs (**d**).

**Figure 8 foods-11-03756-f008:**
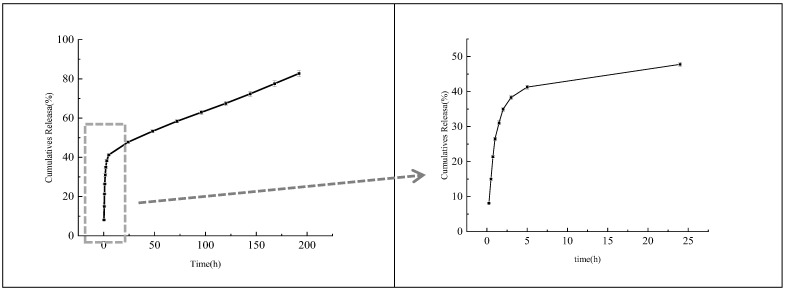
In vitro release profile in saline-phosphate buffer with sodium lauryl sulfate, pH 7.4, from optimized OEO-CSNPs. Data were represented as mean ± SD, *n* = 3.

**Table 1 foods-11-03756-t001:** The code and level of independent variables used in response surface design.

Independent Variables	Code	Level		
		−1	0	1
CS concentration (mg/mL)	A	1	1.5	2
TPP concentration (mg/mL)	B	1	1.5	2
OEO (%)	*C*	0.125	0.25	0.5
Dependent variables	Applied constrains			
*Y*_1_ = %EE	Maximize			
*Y*_2_ = Size (nm)	Minimize			

**Table 2 foods-11-03756-t002:** Coding factor levels and response values for Box–Behnken design.

Run	A Chitosan	B TPP	*C* OEO	*Y*_1_ EE	*Y*_2_ Size
	/mg/mL	/mg/mL	/%	/%	/nm
1	−1	0	−1	81.65	586.23
2	0	1	−1	77.54	286.37
3	0	0	0	91.58	192.33
4	0	0	0	91.94	226.4
5	−1	0	1	85.01	684.37
6	0	1	1	83.41	583.17
7	0	0	0	91.13	188.13
8	−1	−1	0	87.49	515.37
9	1	0	−1	81.91	450.73
10	−1	1	0	81.02	506.67
11	1	1	0	79.99	417.4
12	0	−1	−1	87.47	486.5
13	0	0	0	91.11	186.63
14	0	0	0	92.18	217.17
15	1	−1	0	88.46	283.87
16	0	−1	1	88.56	328.1
17	1	0	1	87.96	375.4

**Table 3 foods-11-03756-t003:** Analysis of variance (ANOVA) of independent variable response surface model.

Source	*Y*_1_-EE%					*Y*_2_-Size				
	Sum of Squares	df	Mean Square	F- Value	*p*-Value Prob > F	Sum of Squares	df	Mean Square	F- Value	*p*-Value Prob > F
Model	348.28	9	38.7	90.23	<0.0001 **	4.03 × 10^5^	9	44,789.02	56.58	<0.0001 **
A	1.24	1	1.24	2.89	0.1328	73,199.03	1	73,199.03	92.47	<0.0001 **
B	112.65	1	112.65	262.67	<0.0001 **	4039.66	1	4039.66	5.1	0.0584
*C*	33.5	1	33.5	78.11	<0.0001 **	3248.58	1	3248.58	4.1	0.0824
AB	1	1	1	2.33	0.1706	5057.34	1	5057.34	6.39	0.0394 *
A*C*	1.81	1	1.81	4.22	0.0791	7522.96	1	7522.96	9.5	0.0177 **
B*C*	5.71	1	5.71	13.32	0.0082 **	51,801.76	1	51,801.76	65.44	<0.0001 **
A^2^	58.59	1	58.59	136.61	<0.0001 **	1.16 × 10^5^	1	1.16 × 10^5^	146.44	<0.0001 **
B^2^	55.11	1	55.11	128.5	<0.0001 **	16,591.9	1	16,591.9	20.96	0.0025 **
*C*^2^	58.43	1	58.43	136.25	<0.0001 **	1.03 × 10^5^	1	1.03 × 10^5^	129.66	<0.0001 **
Residual	3	7	0.43			5540.94	7	791.56		
Lack of Fit	2.09	3	0.7	3.05	0.1546	4193.42	3	1397.81	4.15	0.1014
Pure Error	0.91	4	0.23			1347.52	4	336.88		
Cor Total	351.28	16				4.09 × 10^5^	16			
R^2^	0.9915					0.9864				
Adjusted R^2^	0.9805					0.9690				

Note: * *p* < 0.05 indicates a significant difference; ** *p* < 0.01 indicates a highly significant difference.

**Table 4 foods-11-03756-t004:** Predicted and experimental values obtained from the response optimization of BBD.

Responses	Predicted Value	Experimental Value	Percentage Prediction Error (%)
EE (%)	92.65	92.90 ± 0.35	0.58
Size (nm)	184.83	182.77 ± 4.83	0.78

Results reported are mean ± SD, *n* = 3.

**Table 5 foods-11-03756-t005:** Antimicrobial activities of OEO before and after encapsulated in CSNPs.

	CSNPs	OEO	OEO-CSNPs	Anhydrous Ethanol	Sterile Water
*P. fluorescens*	6.09 ± 0.03 ^Bb^	6.69 ± 0.11 ^Ab^	6.86 ± 0.05 ^Ac^	0.00	0.00
*P. gessardii*	6.63 ± 0.20 ^Ba^	6.07 ± 0.01 ^Cc^	8.30 ± 0.05 ^Ab^	0.00	0.00
*P. jessenii*	6.12 ± 0.02 ^Cb^	8.85 ± 0.10 ^Ba^	9.35 ± 0.04 ^Aa^	0.00	0.00

Results reported are mean ± SD, *n* = 3; values with different letters within the column are statistically different at *p* < 0.05. Uppercase letters indicate differences between bacteria treated differently, whereas lowercase letters indicate differences between bacteria treated similarly.

## Data Availability

Data is contained within the article.

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
