# Peer review of "Preparation, Characterization, In Vitro Release, and Antibacterial Activity of Oregano Essential Oil Chitosan Nanoparticles"

_foods, 2022, doi:10.3390/foods11233756_

Round 1
Reviewer 1 Report
Totally, the research has been well designed. Although, there are some points to review as follows:
- Statistical differences should be added to the table 5.
- Line 448: It is not a "discussion"; it must be a "conclusion" please revise it and don't mentioned many results (details) in this section.
- The discussion can be improved by comparing your result with similar works that studied chitosan nanoparticles loaded with essential oils.
https://doi.org/10.1111/ijfs.15110
https://doi.org/10.1016/j.ijbiomac.2021.11.155
- Proof editing of English language and style required for all the text.
Reviewer 2 Report
in page 3, line 127. What do you mean with cattle?
Dear authors, the main question addressed by the research is the encapsulation of oregano essential oil (OEO) into chitosan nanoparticles (CSNPs). The oregano essential oil chitosan nanoparticles (OEO-CSNPs) were characterized in terms of their particles size, particle size distribution, Zeta potential, and encapsulation efficiency. The chitosan concentration, TPP concentration, and amount of OEO was changed using a BBD design principle, a 3-factor, 3-level complete factorial design. Moreover, encapsulation was followed by FT-IR, and TGA. The morphology was observed by SEM. The antibacterial activity was evaluated before and after encapsulation on three Pseudomonas strains. The objective was to find the optimal conditions for preparing nanoparticles with the highest encapsulation efficiency and the smallest particle size.
Only one comment: in page 3, line 127. What do you mean with cattle?
The topic is original or relevant due to the use of the BBD design principle useful to identify the lowest and highest encapsulation efficiencies and the particle size values range as function of chitosan concentration, TPP concentration, and the amount of OEO added to establish a correlation between the variables used from the experimental results. It fills the gap between experimental and mathematical design because sometimes models are not use in this type of research.
The obtained results by BBD are perfectly explain in terms of the chemical background of the experimental conditions compared to other published materials. The method was very accurate with and error lower than 5%. Moreover, the encapsulation efficiency and release of the OEO is an important parameter that was evaluated as well as the antibacterial activity.
Regarding methodology another stirring speed can be considered, essential oil and emulsifier speed of addition and bacteria evaluated to have a wider range of evaluated variables. Maybe OEO could also be considered as a control.
The conclusions are consistent with the experimental results. The references are adequate, support the article and help to compare the results.
Related to tables and figures: Fig. 1 is adequate to show the formation process of the OEO-CSNPs, the three-dimensional surface representation of the experiment is visually representative, Zeta potential and SEM are also adequate showing the gaussian curve and nanoparticles morphology, respectively. FTIR in Fig. 6 has assigned the wavelengths to the characteristic peak as well as for TGA in Fig. 7. Font in Fig. 8 must be larger. Tables information is adequate.
English does not need revision. I think the novelty of this study is the use of BBD analysis, along with experimental to stablish the ideal conditions for encapsulation and effective release of the OEO.
Reviewer 3 Report
The work by Ma and colleagues is of average originality, the delivery of oregano essential oils with chitosan nanoparticles having been described in numerous other works. The techniques used are also largely established and not particularly innovative, as is the use of a factorial design in the experimental scheme. Overall, however, the study is well executed, clearly written and easy to read. The description of the experimental part is also adequate. I believe, however, that the work needs a major revision in order to be published,
Major
While the part concerning the production and characterisation of nanoparticles is extensively described, in the section concerning antimicrobial assays in my opinion the authors should provide more information, in particular:
Expressing a MIC value would be of greater interest as it would also allow the results to be compared with other similar studies; furthermore, with respect to the test described in the work, I would ask them to add at least some images to support table 5.
Furthermore, I would strongly suggest including time-killing experiments on the bacterial species studied because this would greatly increase the description of the nanoparticles from a functional point of view and would give the work greater interest.
Minor points:
All the figure legends should be more explicative, reporting more information regarding the figure content.
legend of Figure 6 reports "CEO and CEO CSNPs", should it be "OEO and OEO CSNPs" instead ?
Please modify all the bacterial species names in Italic
lines 32 33, the sentence seems incomplete to me
line 36 the sentence is not clear to me, there is a repetion of the word "preservation"?
line 66-67 "The action of polyanions..." is unclear
Round 2
Reviewer 1 Report
The requested revision not carried out completely! A comment has not performed by author in the text!
Discussion! not conclusion! please read the comment again!
"- The discussion can be improved by comparing your result with similar works that studied chitosan nanoparticles loaded with essential oils."
Please revise the manuscript according to the reviewer's comments, if you use any information from other studies, you should mention the articles as references!
